# Utilization of Geogrid and Water Cushion to Reduce the Impact of Nappe Flow and Scouring on the Downstream Side of a Levee

**Fakhar Muhammad Abbas [1] and Norio Tanaka [2,*]**

1   Graduate School of Science and Engineering, Saitama University, Saitama 338-8570, Japan
2   International Institute for Resilient Society, Saitama University, Saitama 338-8570, Japan
*   Correspondence: tanaka01@mail.saitama-u.ac.jp

**Abstract:** Water overflowing from a levee generates scour holes on the toe, which progresses towards the backward crest of the levee and results in nappe flow generation. The direct collision of nappe flow on the downstream area causes levee failure. It is important to introduce a novel countermeasure against scouring caused by nappe flow. Hence, the present study utilized a new technique to reduce scouring due to nappe flow by introducing a combination of pooled water and geogrids. Herein, laboratory experiments were conducted with the three cases for rigid bed (R), named as NR, G1R, G2R (N, G1 and G2 represent no geogrid, geogrid 1 and geogrid 2, respectively), and moveable bed (M), named as NM (nothing moveable), G1M (geogrid 1 moveable), G2M (geogrid 2 moveable), to elucidate the effect of dimensionless pooled water depth ($D_{P*}$), overtopping depth ($D_{C*}$) and the aperture size of geogrids ($d*$) on flow structure and scouring. The results showed that the scour depth was reduced by around 17–31% during the NM cases, 57–78% during the G1M cases and 100% during the G2M cases by increasing the $D_{P*}$ from 0.3 to 0.45. Hence, the combination of geogrids with pooled water (G1M, G2M) performed a vital role in suppressing the scouring, but the results of G2M were more advantageous in terms of scouring countermeasures.

**Keywords:** nappe flow; scouring; water cushion effect; pooled water; geogrid

## 1. Introduction

Levees are important hydraulic structures used to channelize river flows and protect human lives and hinterland areas from disastrous flood inundation. In recent years, flooding has occurred more frequently in Japan, due to global climate change and severe weather conditions [1]. After the devastation caused by the Great East Japan Tsunami (GEJT) in 2011, several cases were documented in which the tsunami travelled up rivers and overflowing occurred from river levees, inflicting the extensive loss of human life and destruction of property [2].

Field surveys conducted by several researchers [3–8] revealed that, following the extensive flooding of the rivers, the levees were breached due to the continuous overtopping of flood water from the levee, despite the flow over the levees being normal in respect to the river flow. When the levee crest is only paved, in the first stage, the continuous overtopping of flood water from the levee causes significant erosive forces at the toe of the levee. These forces further generate scour holes downstream of the levee, and the generated scour holes propagate towards the levee crest. In the second stage of the paved crest levee, the landward slope of the levee is washed away, leaving half of the levee in a safe condition, which generates the nappe flow. This nappe flow further accelerates the scouring and finally causes the breaching of the levee. The whole process of levee failure due to overflowing is presented in Figure 1. So, in the situation described above, the present study considered the second stage, in which the nappe flow was generated with half of the levee structure.

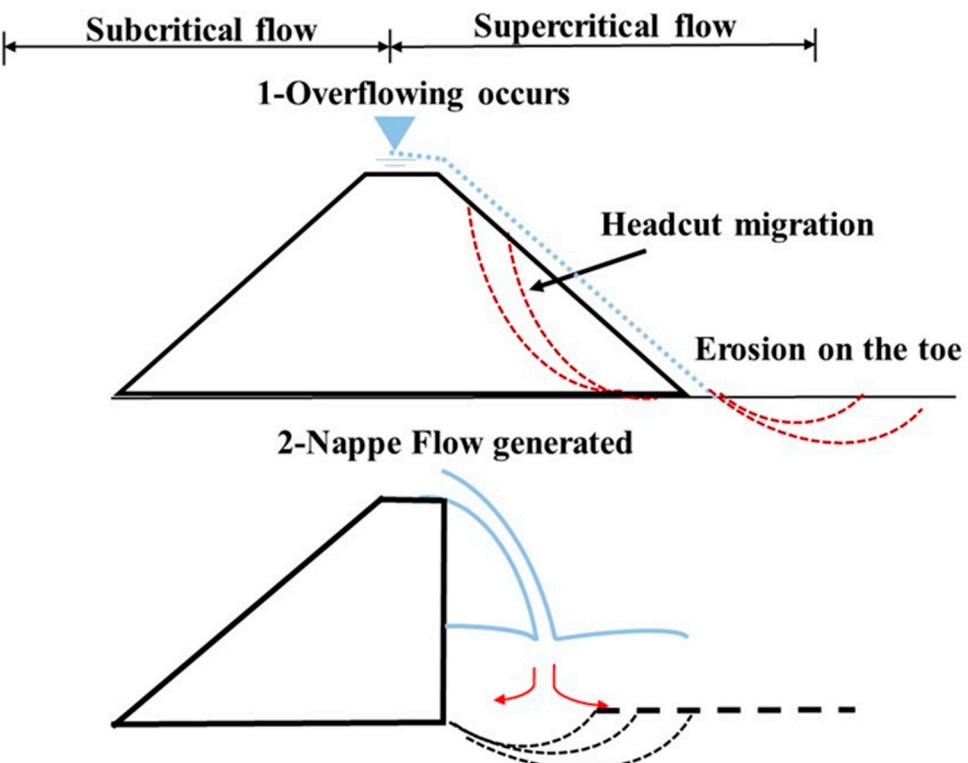

**Figure 1.** Stages of levee failure.

Several studies have been conducted to examine the scour development process around hydraulic structures, such as piers [9–11], spur dikes [12,13], weirs [14] and embankments/levees [15–18], and to suggest alternative countermeasure techniques to decrease scouring around them [19–22]. For example, Muhammad and Tanaka [23] revealed in their experimental study that the toe of an embankment can be protected from excessive scouring and the energy of overflowing water can be reduced by installing a double-layer dense coastal vegetation structure behind the embankment. On the other hand, they argued that in practice, dense forests are difficult to build because they require large spaces to grow. Furthermore, one of the most important lessons learnt from the 2011 Great East Japan Tsunami (GEJT) is that as a tsunami defense strategy, depending on coastal vegetation alone [24–26] is inadequate. Zaha et al. [27], Kamiwada et al. [28] and Abbas and Tanaka [29] utilized a moat structure behind an embankment, which significantly reduced the fluid force, overflow volume and energy, respectively. Another option is the utilization of concrete blocks to protect the area from local scouring [30–32]. However, despite the efficacy of concrete blocks in terms of minimizing scouring and reducing energy, the concrete blocks also caused the scour hole to be relocated farther downstream of the embankment. In addition, a large amount of money must be spent on digging and transporting the heavy materials required to construct the concrete blocks.

Recently, with the development of geosynthetics, materials such as geogrids have been widely utilized in the prevention of soil erosion for stability on embankments and highways [33,34]. The construction of levees or embankments is vulnerable to erosion from continuous surface water runoff when the slope is not stabilized and erosion is not controlled. An experimental investigation was carried out by Takegawa et al. [35] who used geogrids over the gravel bed, which protect the toe of the levee. They reported that the geogrid prevented local scouring to some extent, but the high-energy overflowing water shifted the scour hole farther downstream, which was especially detrimental to downstream buildings. However, due to its light weight nature, water permeability, and low cost, the usage of geogrids as a countermeasure proved to be quite useful. Therefore,

both the energy and scouring caused by overflowing water had to be reduced, which the geogrid alone was unable to do.

Another research conducted by Tanaka and Sato [6] reported that the scour holes were created behind the overflowing embankments during the 2011 tsunami (GEJT) were proven to be the most effective means of mitigating erosion due to the pooling of water in these holes. These scour holes and excavated ground behind the levee were also helpful in lowering the energy of tsunamis or floods [5,27,36,37]. Similarly, a water cushion or pool water located at a stilling basin's foundation is given protection by reducing the energy and velocity of the flow [38]. Hence, based on the research mentioned above, it was necessary to utilize the water cushion/pooled water and geogrid in conjunction as a new countermeasure against the destruction produced by overflowing nappe flow and levee breaching, which has not yet been clarified.

As a result, the primary goal of this work was to develop a new countermeasure technique by combining pooled water and geogrids to alleviate the downstream effects of overflowing nappe flow and scour. Firstly, several pooled water depths were selected on the assumption that the accumulated pooled water would operate as a water cushion that would counteract the approaching nappe flow, as indicated by [39]. Then, the combination of pooled water and two different geogrids with large (G1 = 6.5 mm) and small (G2 = 2.5 mm) aperture sizes were utilized to investigate their combined effect on the impact of nappe flow and scour behaviour on the downstream side of the levee structure.

Finally, this research proposes which type of geogrid and pooled water, or water cushion, is appropriate to reduce the impact of the overflowing nappe flow and associated scouring on the downstream area.

## 2. Materials and Methods

### 2.1. Flow Conditions

The experiments were conducted in a 16 m long, 0.5 m wide, and 0.7 m deep rectangular open channel laboratory flume at Saitama University, Japan. Figure 2a represents the laboratory flume and experimental setup. During the Great East Japan Tsunami (GEJT) of 2011, the Abukumagawa river levee in Japan's Tohoku area experienced overtopping depths ranging from 0.84 to 1.78 m [6]. Similarly, during the post tsunami survey conducted by Tokida and Tanimoto [36], overtopping depths between 2.8 and 6.8 m were reported. Since a real flood wave has a very lengthy duration, its regeneration in an experimental setting is difficult and unsuitable for sufficient scaling, while the flood flow in the flume could be reproduced by a control pump discharge ($m^3$/s) [40]. Floods waves are often referred to as quasi-steady flows because of their extended duration. Therefore, to guarantee an adequate scale on a 1/100 scale, pump discharge ($m^3$/s) was employed to create the flood wave in this experiment. The flow was maintained constant by a valve and a flow meter (Signet 8150 flow totalizer). To simulate the real flood flow, all experimental trails were set to a model scale of 1/100 and three quasi-steady critical overflow flow depth conditions were selected, ranging from ($D_c$ = **2–4 cm**). The range of critical overflow water depths selected was between the lowest and highest overflow depths reported in the previous study [36]. Froude similarity was employed. Several discharge trials were conducted, before deciding on the range of flow conditions to be evaluated. Water depths were measured against each discharge number with a rail-mounted point gauge (KENEK corporation, accuracy $\pm$ 0.1 cm) and used to calculate the corresponding Froude number ($Fr_o$) on the levee's brink. A total of three Froude numbers, ranging from 1 to 1.3 were derived from the three critical overflow depths ($D_c$ = **2, 3, 4 cm**) measured at the crest of the levee and discharge values, respectively.

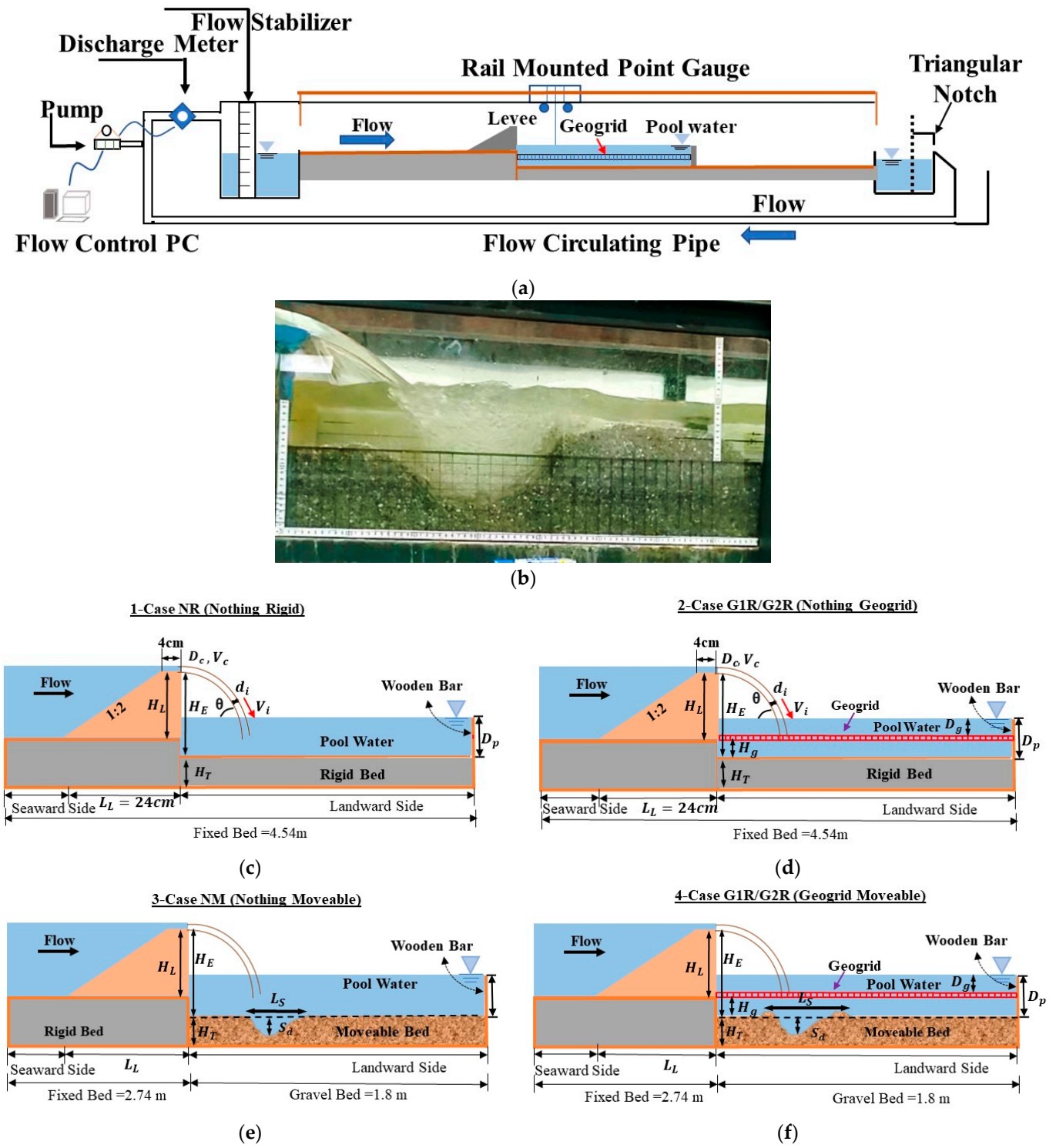

**Figure 2.** (**a**) Schematics of experimental setup, (**b**) picture of experimental setup. (**c**) Test section for nothing rigid bed (NR) experiments, (**d**) for geogrid rigid bed (G1R/G2R) experiments, (**e**) for nothing moveable bed (NM) experiments and (**f**) for geogrid moveable bed (G1M/G2M) experiments.

### 2.2. Model Features

The levee and pooled water/water cushion were used as the models during the experiments. As discussed in the introduction section, we focused on the second condition in which the half of the paved crest levee was breached or washed out due to continuous overflowing of water and nappe flow was generated. As a result, the river side of the

levee consisted of a slope with a ratio of (1:2), while the landward side was a straight vertical. In addition to this, the length of the levee ($H_L$) was originally 44 cm but due to the beaching of the levee from its landward slope, the length of the levee was taken as 24 cm, including the river side slope (20 cm) and the crest (4 cm), as shown in Figure 2c–f. Following the 2011 tsunami (GEJT), the Japanese government planned to elevate and/or replace the embankments/levees in Fukushima, Miyagi, and Iwate prefectures to a height of (7.25–14.7) meters in order to decrease the effect of flooding due to level 2 tsunamis on these three regions [21,41]. Therefore, in this study, on a 1/100 scale, the levee was scaled down to a height ($H_L$) of 10 cm (10 m on a real scale), whereas the height of the nappe flow that strikes the downstream bed from the vertical end of the levee, which was denoted with a notation ($H_E$), was considered to be 20 cm, as shown in Figure 2c–f. A pool/water cushion with a length of 1.8 m and a width of 0.5 m was placed directly downstream of the levee model, whereas the four pool heights ($D_P$) were selected ranging from 6 to 9 cm. An adjustable wooden board was positioned at the end of the pool to achieve the desired pooled water depths ($D_P$), and the pool was filled with water before beginning the experiment trail. Two geogrids with large (G1–6.5 mm) and small (G2 = 2.5 mm) mesh sizes were selected [35]. These geogrids were installed separately inside the pooled water at a fixed height ($H_g$) of 5 cm from the downstream channel's bed. In addition, the depth of the water ($D_g$) from the geogrid height to the surface of the polled water was altered depending on the selected pooled water depths ($D_P$).

*2.3. Rigid and Moveable Bed Conditions*

Two phases of experiments were conducted in this present study. During the first phase, tests were carried out in a pool with a rigid bed (R) composed of wooden material to investigate the combined effect of varying pooled water depths and geogrids on the flow structure. A total of 36 experimental trails (Nos. 1–36 in Table 1) were conducted under the fixed bed conditions. The cases with only pooled water with rigid beds were named as NR (where N represents nothing, i.e., without a geogrid, and R represents rigid bed), as shown in Figure 2b, while the cases with a combination of pooled water rigid beds and geogrids were named as G1R and G2R (where G1 depicts geogrid-1 with an aperture size of 6.5 mm, and G2 represents geogrid-2 with an aperture size of 2.5 mm; R represents rigid bed), as shown in Figure 2c.

**Table 1.** Hydraulics and geometric conditions.

| Trial No. | Case Name | Geogrid | | Dimensionless Overtopping Depth ($D_{C*}$) | Dimensionless Pool Water Depth ($D_{P*}$) |
|---|---|---|---|---|---|
| | | Aperture Size *d* (mm) | Shape of Mesh | | |
| 1–12 | NR | - | circle | 0.1, 0.15, 0.2 | 0.30, 0.35, 0.40, 0.45 |
| 13–24 | G1R | 2.5 | circle | 0.1, 0.15, 0.2 | 0.30, 0.35, 0.40, 0.45 |
| 25–36 | G2R | 6.5 | circle | 0.1, 0.15, 0.2 | 0.30, 0.35, 0.40, 0.45 |
| 37–48 | NM | - | circle | 0.1, 0.15, 0.2 | 0.30, 0.35, 0.40, 0.45 |
| 49–60 | G1M | 2.5 | circle | 0.1, 0.15, 0.2 | 0.30, 0.35, 0.40, 0.45 |
| 61–72 | G2M | 6.5 | circle | 0.1, 0.15, 0.2 | 0.30, 0.35, 0.40, 0.45 |

During the second phase, a total of 36 experimental trails (37–72 in Table 1) were conducted under the moveable bed conditions (with nothing/no geogrid with moveable bed (NM), as shown in Figure 2d, and geogrid with moveable bed (G1M/G2M), as shown in Figure 2e) to investigate the role of various pooled water depths and geogrids on scouring behaviour. An aggregate with a dry density of 2650 kg/m$^3$ and a median grain size ($d_{50}$) of 4.47 mm was chosen for the bed materials as a soil layer [20]. Local souring occurred in 2011 GEJT along dikes/levees and trees due to a greater threshold value for scouring, caused by the bed's cohesive soil composition. Numerous authors have performed laboratory studies in clear water scouring conditions, using a grain size of 4.5 mm as a bed material for the soil layer to achieve optimum scouring [42,43]. Furthermore, since the soil, in general, is very cohesive, when employing non-cohesive particles as soil, the particle size must

be greater with higher gravity to reflect soil cohesivity [44]. Otherwise, the cohesive soil material would be entirely washed away, due to the direct impact of the high intensity overflow depth on the bed. Therefore, based on the above reasons, the non-cohesive grain size or diameter of the particle used for the movable bed material was selected to ensure the scouring phenomenon under clear water conditions [9,10,26,28,35,45], for which the Froude similitude law is not applicable. Thus, the movable bed was filled with gravel particles and placed from the levee edge to farther downstream of the pooled water edge, covering an area of 1.8 m long, 0.25 m in width and a constant height ($H_T$) of 0.15 m. As a support, a wooden flat plate with a depth equal to the gravel bed was set at the pooled water edge to protect the gravel layer from washing away during the flow.

*2.4. Flow Duration*

The scouring phenomenon caused by overflowing is primarily based upon the grain size of the bed material, i.e., (cohesive or non-cohesive material) when it comes to the interaction between the bed and the soil layer [46]. Since cohesive soil layers in model size studies may be quite challenging to employ, many researchers, as mentioned above (Section 2.3), have used non-cohesive grains as a soil layer in the context of scouring investigations. Another critical consideration is the flow duration for the development of the scour profile for a certain particle size. Considering the non-cohesive particle size as a bed material, maximum scouring can be estimated when the scouring profile achieves an equilibrium state [35]. Hence, in this present study, the flow duration was chosen when the scour depths achieved the equilibrium state. Figure 3 shows the development of scouring ($S_d$, maximum scour depth) as a function of overflow time (in seconds) for case NM (nothing/without geogrid moveable) against the maximum dimensionless overtopping depth ($D_{p^*} = 0.4$). It is obvious from the Figure 3 that once the overflow time (from the levee model) reached about 200 s (approx., 3 min 20 s), the equilibrium scoured depth could be attained. Thus, this flow duration was chosen for all the experimental trails because the highest scouring depth achieved equilibrium conditions at the overflow time of 200 s, and no significant changes were observed in the scouring profiles after this flow duration. Moreover, according to the Froude similitude rule of the physical scale, it was probably 1600 s (33 min) on the real physical scale of 1/100.

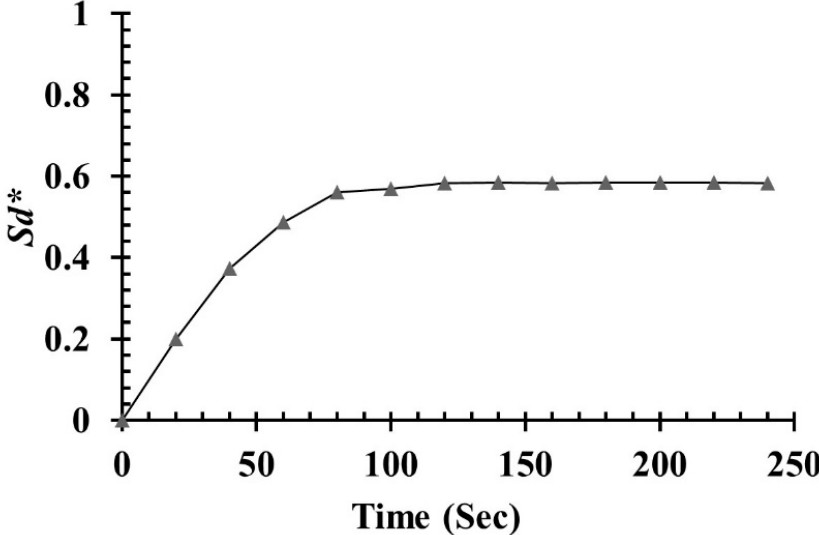

**Figure 3.** Time progress of dimensionless maximum scour depth ($S_d$*) at the initial stage to set the equilibrium state.

*2.5. Acquisition System of the Experimental Data*

A previous study conducted by [10] demonstrated that the ratio of the channel width to the approaching flow depth ratio should be greater than 5 to limit the influence of

side walls; therefore, this ratio was maintained in the present study. Moreover, to obtain accurate experimental data, prior research conducted by Anjum, and Tanaka [47] suggested that, to reduce the impact of the secondary circulation induced by side walls on the water surface profiles, the water depth measurement should be taken at the centre of the channel. Therefore, in the present study, during the 1st phase, i.e., only rigid bed with nothing (NR) and geogrid cases (G1R/G2R), the water levels were measured throughout the centre of the channel flume by using the rail mounted point gauge (KENEK corporation, accuracy ± 0.1 cm). Moreover, to observe the types of the flow structures caused by the hydraulic jump, the flow behaviour was visualized with a high definition (HD) camera (model = Olympus, frame rate = 1/30). Overall, two different video cameras were used, which were placed at the top and side of the channel, respectively. These HD cameras were operated with the computer running photography software K-II ver.1.03 to record and analyze the videos. In each experimental trail, the water levels were analyzed (within the pooled water/water cushion) through the captured videos. Because of the complicated flow patterns and fluctuations inside the pooled water, the minimum and maximum depths of the water surface were noted at a fixed position with intervals of 1.8 cm to 5 cm in various consecutive frames and then averaged. During the 2nd phase, i.e., moveable bed with nothing/no geogrid (NM) and geogrid cases (G1M/G2M), the same (HD) camera was used to visualize the scour development process. After each experimental trail, the three-dimensional (3D) laser displacement gauge (model name = Keyence LK-500 cooperation) was used to measure the scouring at a small interval of 1–2 cm (for obtaining the accuracy), depending on the variation in the gravel bed both in the longitudinal and transverse directions. The 3D laser displacement gauge was connected to the PC, and the LJ navigator was used to obtain the data of the scour profile, which was further analyzed in Fortran Software to obtain the final values of the scour depths. Finally, the scouring profiles were created using the measured scouring data.

### 2.6. Dimensional Analysis

It is mandatory to scale down the prototype model to a smaller size to investigate and observe the actual phenomena in laboratory experiments under controlled conditions. In the present experimental study, the scale of 1/100 was considered to estimate the behavior of the prototype. This study mainly focused on the impact of nappe flow and scouring on the downstream side by considering the following associated dimensional parameters (1), as shown in Figure 2b–e.

$$f_1\left(D_C,\ D_P, D_g,\ H_E,\ H_T,\ S_d,\ d,\ d_i,\ L_S, V_c,\ g, \rho, \mu, \sigma\right) = 0 \tag{1}$$

where $D_C$ represents the critical overflow depth; $D_P$ is the pooled water depth; $H_g$ is the height of the geogrid above the bed (5 cm); $D_g$ is the flow depth above the geogrid ($=D_P - H_g$); $H_E$ is the height of the embankment from the downstream toe, which was constant, i.e., 20 cm; $H_T$ = height of moveable/rigid bed, which was taken as constant, i.e., 15 cm; $S_d$ is the maximum scour depth; $d$ is the aperture size of the geogrid; $d_i$ is the nappe thickness, $L_s$ is the length of the scour hole; $V_c$ is the initial critical flow velocity, $g$ is the gravitational constant, $\rho$ is the water density in kg/m$^3$, and $\mu$ is the viscosity of the water.

The Buckingham's pi theorem was used to derive the following modified Pi groups to identify the relevant dimensionless parameters and the governing ones are considered in the results:

$$f\left(\frac{D_C}{H_E},\ \frac{D_P}{H_E},\ \frac{d}{di},\ \frac{S_d}{H_T},\ \frac{L_s}{H_T},\ \frac{V_c}{\sqrt{gD_c}},\ \frac{\rho D_c V_c}{\mu},\ \frac{\rho D_c V_c^2}{\sigma}\right) = 0 \tag{2}$$

$$f(D_{C^*},\ D_{P^*},\ d^*,\ H_E,\ H_T,\ S_{d^*},\ L_{S^*},\ Fr_o,\ Re,\ We) = 0 \tag{3}$$

Water density and dynamic viscosity, as well as air–water surface tension, are important fluid characteristics to consider in free-surface gravity flow. As a result, it is necessary to maintain the same scale parameters as the actual experiment, such as Froude, Reynolds, and Weber numbers, to validate this model scale experiment. It is common practice to

use Froude similarity rather than the Reynolds number to describe free-surface gravity flows [48]. Therefore, to specify the hydraulic conditions, Froude similarity was employed in the current study (discussed in Section 2.1). Furthermore, in each case, the density and viscosity of the water were the same; therefore, the Reynolds number was ignored.

The experiments were carried out in a smooth glass sided wall with a very small bed slope and the water level was measured along the centerline of the channel. Thus, the influence of wall roughness was also eliminated. Furthermore, to reduce the effect of water surface tension, which is related to the dimensionless Weber number, the lowest necessary Weber number should be greater than 11 [49]. Previous research conducted by Peakall and Warburton [50] has also shown that the Weber number threshold ranges from around 10 to 120, and values less than this result in some surface tension-induced deformation in the modelled experiment. The Weber number determined in this investigation was larger than 10 and varied between 67 and 267; thus, it was supposed that the surface tension had no effects on the trails and was thought to be negligible.

This study mainly focused on the scour reduction in the flood overtopping flow through the combination of pooled water and geogrids; therefore, the scour reduction rate, i.e., scour depth ($S_{d*} = \frac{S_d}{H_T}$) and scour length ($L_{S*} = \frac{L_s}{H_T}$), is a function of the dimensionless overtopping depth ($D_{c*} = \frac{D_C}{H_E}$), in which $D_C$ is calculated from the Froude similarity (discussed in Section 2.1), pooled water depths ($D_{p*} = \frac{D_P}{H_E}$), and aperture size of the geogrid $\left(d^* = \frac{d}{di}\right)$.

$$S_{d*}, L_{S*} = f(D_{C*}, D_{P*}, d^*) \tag{4}$$

Hence, three varying dimensionless overtopping depths ($D_{c*} = \frac{D_C}{H_E}$, (0.1, 0.15, 0.2), in which $D_C$ was considered as 2, 3, and 4 cm and $H_E$ was constant, i.e., 20 cm), and four different dimensionless pooled water depths ($D_{p*} = \frac{D_P}{H_E}$, 0.30, 0.35, 0.40, 0.45, in which $D_P$ was selected as 6, 7, 8 and 9 cm and $H_E$ was constant i.e., 20 cm) were chosen for the experimental trials. The geogrids with two different aperture sizes, ($d$) = (G1) 6.5 mm and (G2) 2.5 mm, were selected.

### 2.7. Determination of Jet Thickness, Angle with Horizontal and Impact Velocity

The relation between overtopping depth ($D_C$) and thickness ($d_i$), impact velocity ($V_i$) and angle ($\theta$) of the approaching nappe is given in Table 2. Rand [51] conducted several experiments and presented the following Equations (5)–(7) to find the thickness, velocity, and angle of the approaching nappe flow.

$$d_i = D_c * 0.687 * \left[\frac{D_c}{h}\right]^{0.483} \tag{5}$$

$$V_i = V_c * 1.455 * \left[\frac{D_c}{h}\right]^{-0.483} \tag{6}$$

$$tan\theta = 0.838 * \left[\frac{D_c}{h}\right]^{-0.586} \tag{7}$$

where $d_i$ is the nappe thickness, $D_C$ is the critical overflow depth, $h$ is the height of the levee, $V_i$ is the velocity of the approaching nappe, $V_C$ is the critical velocity, $\theta$ is the angle of the approaching nappe with a horizontal layer. Table 2 indicates that the magnitude of $V_i$ and $d_i$ increases with increasing $D_C$, while the angle ($\theta$) decreases with increasing $D_C$.

Furthermore, Yen [52] and Agostino and Ferro [53] presented an analytical approach to find the maximum scour depth ($S_d$) and length of scour depth ($L_s$), respectively. The experimental results of $S_d$ and $L_s$ were compared with the results.

**Table 2.** Analysis of experiments.

| $D_{C^*}$ | Thickness of Approaching Nappe ($d_i$) m | | Angle with Horizontal ($\theta$) | | Impact Velocity |
| --- | --- | --- | --- | --- | --- |
| | Experimental | Analytical (Equation (5)) | Experimental | Analytical (Equation (7)) | ($V_i$) m/s (Equation (6)) |
| 0.1 | 0.005 | 0.0055 | 70.86 | 71.63 | 1.95 |
| 0.15 | 0.0089 | 0.008 | 69.1 | 68.45 | 1.96 |
| 0.2 | 0.0133 | 0.0123 | 66.34 | 65.27 | 1.98 |

## 3. Analysis of Results and Discussions

### 3.1. Flow Structure Analysis

3.1.1. Flow Structure Manifestation and Its Physical Identification

The flow structure was mainly classified upon the base of different types of hydraulic jump. It was noticed during all cases that the formation of hydraulic jump was mainly dependent upon the dimensionless overtopping depth ($D_{c^*}$), and aperture size of the geogrid ($d^*$), irrespective of the dimensionless pooled water depth ($D_{p^*}$). During all the experimental trials, overall, four manifestations of hydraulic jump were observed and were named as type-(a, b, c and d), respectively, as shown in Figure 4.

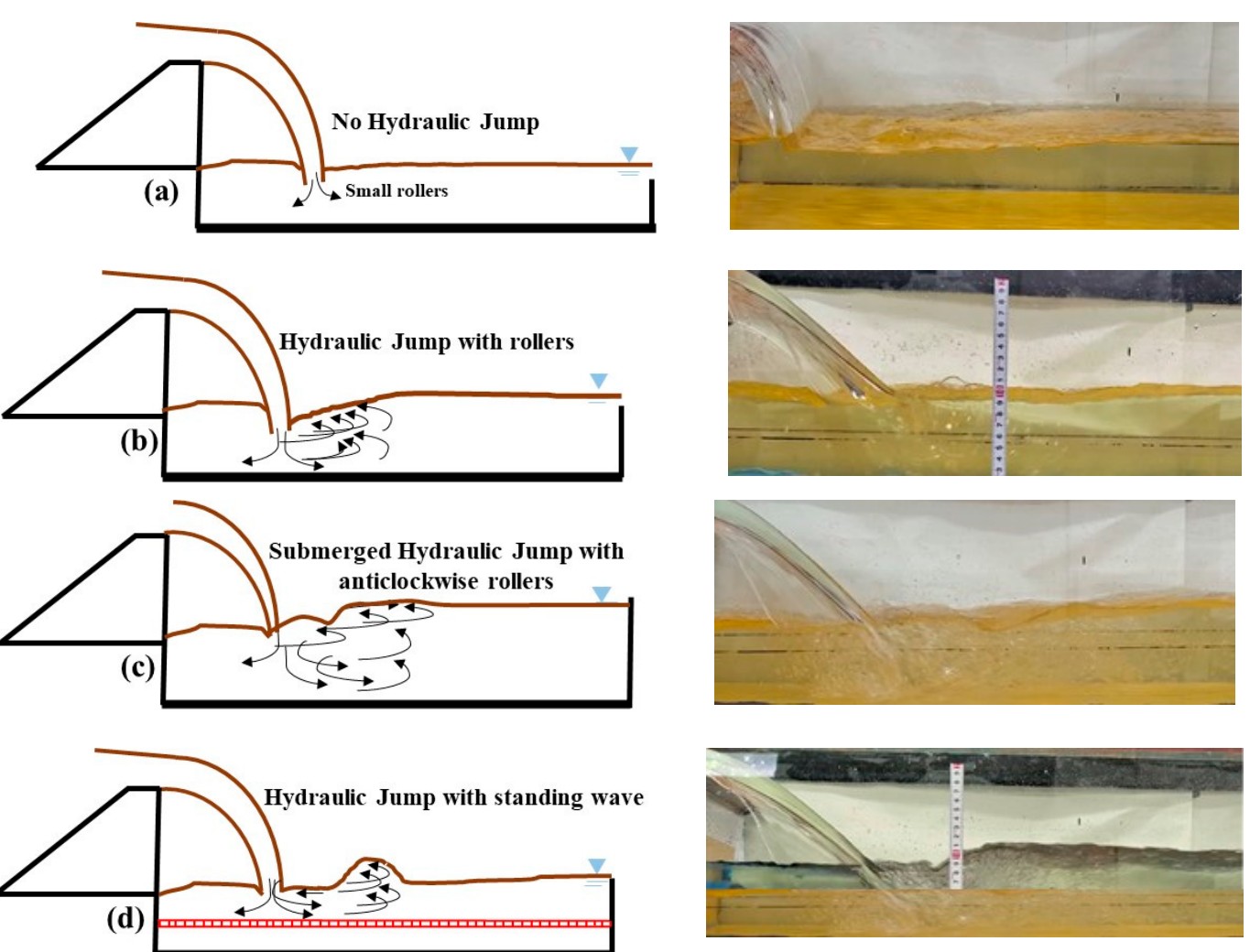

**Figure 4.** Manifestation of observed flow structure. (**a**) Type (a); (**b**) type (b); (**c**) type (c); (**d**) type (d).

a.  No hydraulic jump was observed.
b.  After nappe impinged on the pooled water, hydraulic jump was generated with a distinct surface roller.

c.　　A submerged hydraulic jump was formed.

d.　　Hydraulic jump with a standing wave was generated.

### 3.1.2. Flow Structure Classification

Table 3 represents that the flow structure classification for the NR, G1R and G2R cases, at various dimensionless overtopping depths ($D_{c*}$), pooled water depths ($D_{p*}$) and aperture sizes ($d^*$) of the geogrid. During all cases (NR, G1R and G2R), the flow structure was reported as type (a) at all pooled water depths ($D_{p*}$ = 0.30, 0.35, 0.40 and 0.45) and at the lowest $D_{c*}$ = 0.1. For type (a), the nappe impinged on the pooled water with lower velocity and there was no evidence of hydraulic jump, as shown in Figure 4a. As the overtopping depth was raised ($D_{c*}$ = 0.15, 0.2), the velocity and thickness of the approaching nappe also increased, causing the impact of the approaching nappe to increase, converting the flow structure into type (b) and (c), as shown in Figure 4 during the NR and G1R case. Markel et al. [54] observed similar flow manifestation. During these flow structures of types (b) (Figure 4b) and (c) (Figure 4c), the impinging nappe flow directly struck the pooled water up to the bed, where it bounced upstream and generated distinct surface rollers and hydraulic jump. For maximum $D_{c*}$ = 0.2, only type (c) was observed, where the approaching nappe flow appeared in two parts. The first part bounced upstream and joined the mainstream flow and second part was reflected, which generated the anticlockwise rollers and tended to form a complete submerged hydraulic jump, as shown in Figure 4c. It can be said that the impact of the approaching nappe flow significantly varies with the variation in the dimensionless overtopping depth $D_{c*}$ during the NR and G1R cases, irrespective of the dimensionless pooled water depth $D_{p*}$.

**Table 3.** Flow structure classification.

| Case Name | $D_{P*}$ | $D_{C*}$ = 0.1 | $D_{C*}$ = 0.15 | $D_{C*}$ = 0.2 |
|---|---|---|---|---|
| NR cases | 0.45 | a | b | c |
| | 0.4 | a | b | c |
| | 0.35 | a | b | c |
| | 0.30 | a | b | c |
| G1R cases | 0.45 | a | b | c |
| | 0.4 | a | b | c |
| | 0.35 | a | b | c |
| | 0.30 | a | b | c |
| G2R cases | 0.45 | a | d | d |
| | 0.4 | a | d | d |
| | 0.35 | a | d | d |
| | 0.30 | a | d | d |

The manifestation of flow structure significantly varied after applying geogrid-2 (G2 with aperture 2.5 mm) during the G2R cases, as compared to the NR and G1R cases. Initially, the flow structure was type (a) at the lowest $D_{c*}$ = 0.1, while it was converted into type (d) as the $D_{c*}$ was increased from 0.1 to 0.15 and 0.2, respectively, during all the $D_{p*}$. The G2 played an important role during the formation of type (d), as shown in Figure 4d, because when the approaching nappe flow impinged on the pooled water, it faced two obstacles. The first one is the pooled water, which provides a counterforce and acts as a flexible bottom plate [39]. The second obstacle was the geogrid (G2) with fine meshing, which acts as a rigid bed that generates hydraulic jump with a standing wave. On the contrary, Markel et al. [54] observed this type (d) of flow structure at the highest $D_{c*}$ and lowest $D_{p*}$, where the approaching flow had to face a lesser water cushion effect due to the lower pooled water depth and it directly struck the downstream bed and generated hydraulic jump with a standing wave. The presence of G2 with the pooled water depth proved to be very advantageous to minimize the impact of the approaching nappe flow, which may protect the downstream bed against erosion.

It can be said that the flow structure during NR and G1R was mainly dependent upon the $D_{c^*}$, irrespective of the $D_{p^*}$ and $d^*$. On the contrary, the flow structure significantly varied due to the $D_{c^*}$, in addition to the aperture size of the geogrid ($d^*$) during the G2R cases. From the above discussion, it can be concluded that the G2R cases proved very effective in changing the flow structure and in minimizing the impact of the approaching nappe flow, as compared to the NR and G1R cases.

### 3.2. Analysis of Scouring

3.2.1. Effect Dimensionless Pooled Water Depth ($D_{p^*}$) and Overtopping Depth ($D_{c^*}$) on Scour Profile

A variation in scouring due to the nappe flow was observed for all the cases against the varying non dimensional overtopping depths ($D_{c^*}$) and pooled water depths ($D_{p^*}$). During all the cases, including NM, G1M and G2M, the scouring is initiated at the place where the nappe flow directly strikes, and it propagates further both upstream and downstream and finally, a scour hole is generated, as shown in Figures 5–7, respectively. The scour depths were increased gradually and after some time, they reached their maximum value and later became uniform (equilibrium state). It can be observed from Figures 5–7 that the scour profiles significantly varied after changing the $D_{c^*}$, $D_{p^*}$ and $d^*$, respectively. Figure 5a–c represent the scour profiles variations after changing the $D_{p^*}$, and $D_{c^*}$ during the NM cases. During the NM cases, the scour phenomena were mainly dependent upon two considered parameters, i.e., $D_{c^*}$ and $D_{p^*}$.

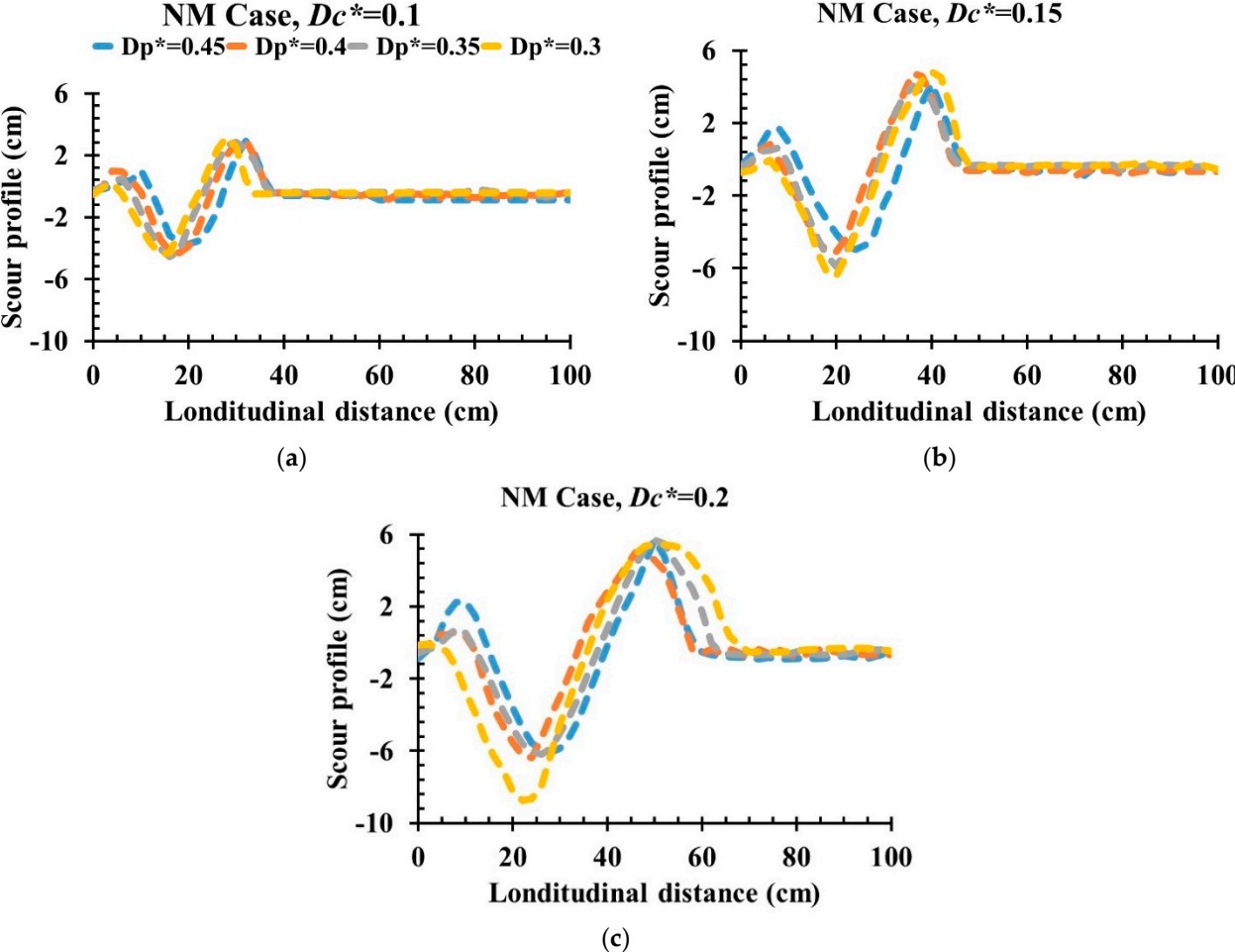

**Figure 5.** (**a**–**c**) Scour profile during NM case (nothing case): (**a**) during $D_{c^*}$ = 0.1; (**b**) during $D_{c^*}$ = 0.15; (**c**) during $D_{c^*}$ = 0.2.

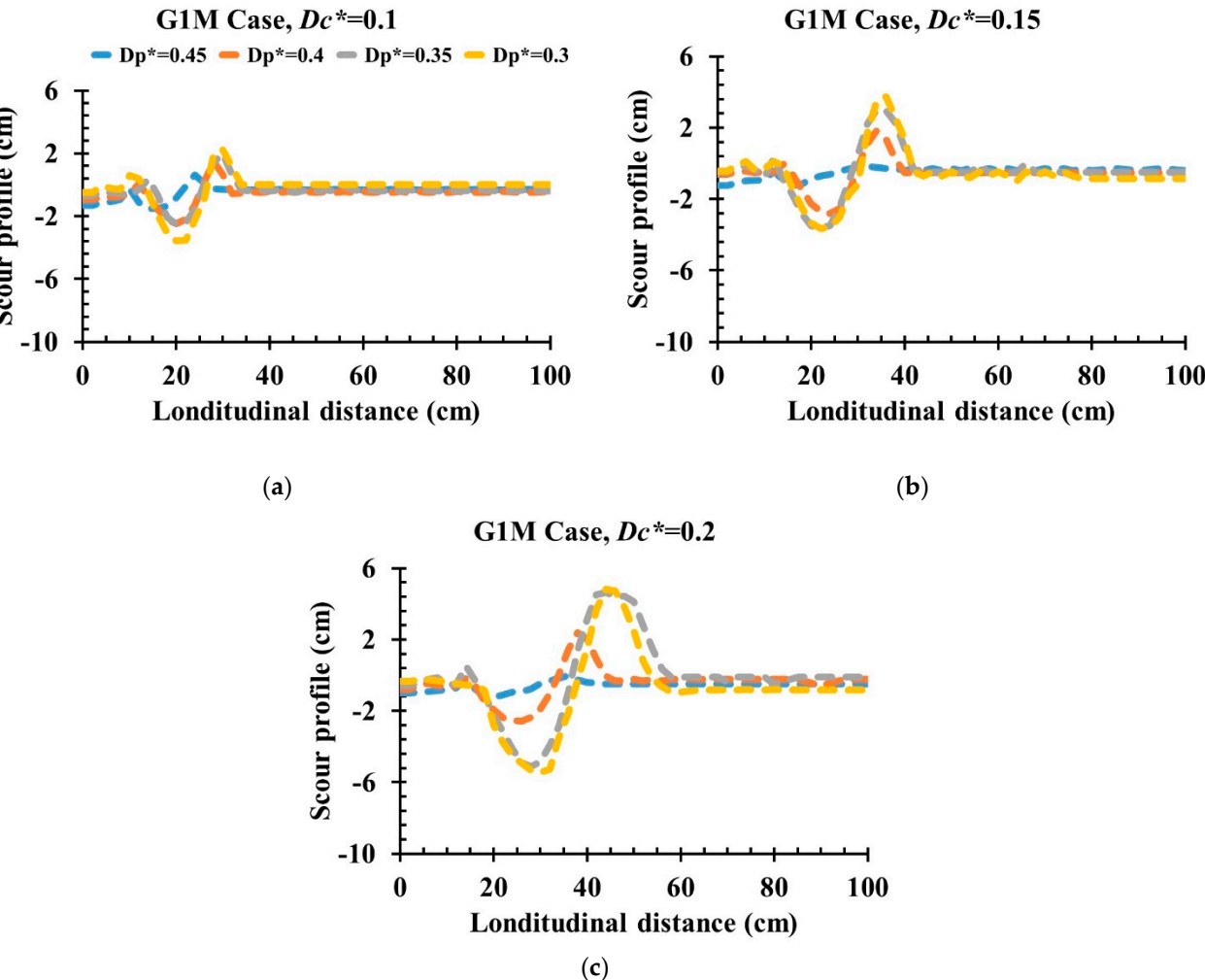

**Figure 6.** (**a**–**c**) Scour profile during G1M case: (geogrid.1 case with aperture 6.5 mm): (**a**) $D_{c*} = 0.1$; (**b**) $D_{c*} = 0.15$; (**c**) $D_{c*} = 0.2$.

As it can be observed from Figures 5a and 8a,b, the maximum scour depth of the hole and its length reached their minimum values at the lowest dimensionless overtopping depth ($D_{c*} = 0.1$) during all the pooled water depths, While the scour depth and length increased with the increase in ($D_{c*} = 0.15$) and reached their maximum values at the highest ($D_{c*} = 0.20$), as shown in Figure 5b,c and Figure 8a,b. It was due to this reason that increasing the $D_{c*}$ increased the impact velocity and thickness of the approaching nappe flow, as shown in Table 2. Afreen et al. [43] and Takegawa et al. [35] also reported similar results during their experiments and stated that the scour depth increased with the increase in $D_{c*}$. The experimental results of the maximum non dimensional scour depth ($S_{d*}$) and length of scour hole ($L_{s*}$) have been compared with the analytical results shown in Figure 8a,b. It has been observed that $S_{d*}$ and $L_{s*}$ were very close to the results of the experimental trial with the lowest $D_{p*}$ at all $D_{c*}$.

The scour profiles show that the scour depth decreases with the increasing $D_{p*}$ during all NM cases, as shown in Figures 5a–c and 8a,b, respectively. Approximately, scour reduction rates of 32%, 23% and 18% were observed by changing the $D_{p*}$ from 0.30 to 0.45 when $D_{c*} = 0.1$, 0.15 and 0.2, respectively. The scour reduction with the increments in $D_{p*}$ was due to the water cushion effect of the accumulated pooled water, which applies a counterforce to the directly striking nappe flow [38], which reduced the impact of the nappe flow.

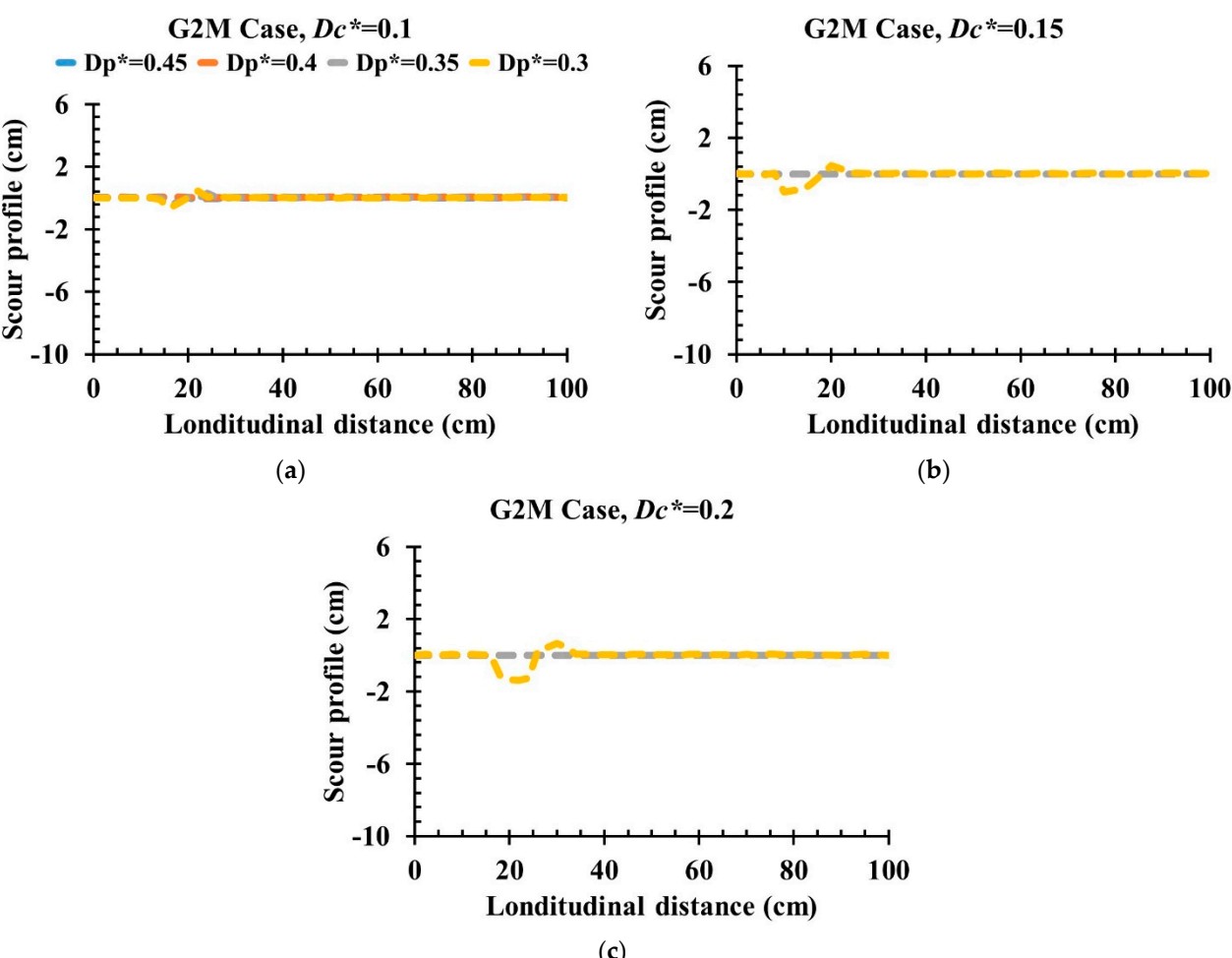

**Figure 7.** (**a**–**c**) Scour profile during G2M case: (geogrid.2 case with aperture 2.5 mm): (**a**) $D_{c^*} = 0.1$; (**b**) $D_{c^*} = 0.15$; (**c**) $D_{c^*} = 0.2$.

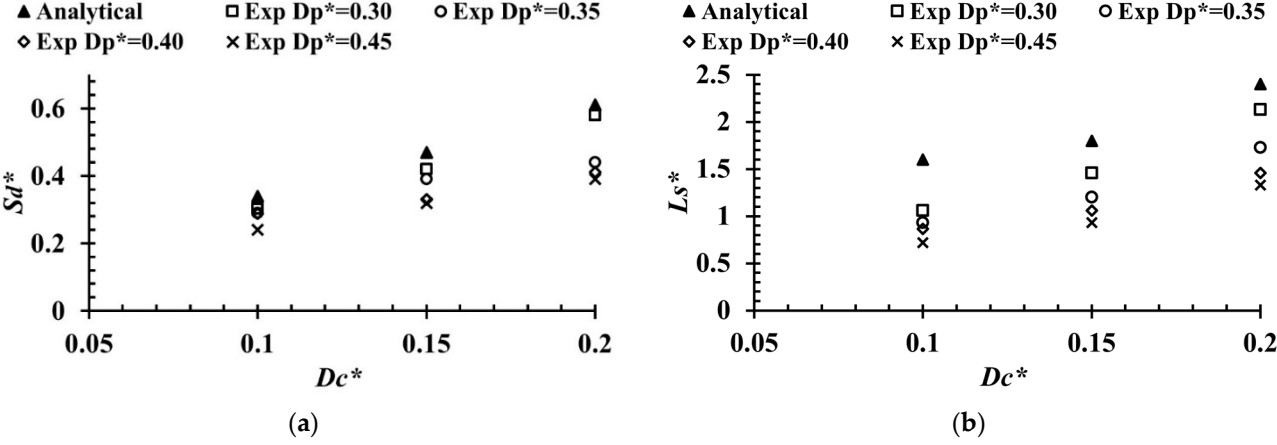

**Figure 8.** (**a**) Relation between maximum scour depth ($S_d^*$) and (**b**) length ($L_{S^*}$) and dimensionless overtopping depth ($D_{c^*}$) during NM case.

### 3.2.2. Combined Effect of Dimensionless Pooled Water Depth ($D_{p^*}$) and Geogrid on Scour Profile

The scour phenomena were significantly affected by applying the combination of pooled water and the geogrid. Figures 6a–c and 7a–c represent the scour profiles during the G1M and G2M cases for the dimensionless overtopping depths ($D_{c^*}$) of 0.1, 0.15 and 0.2,

respectively. It can be observed from the scour profiles (Figures 6 and 7) that the utilization of pooled water depths ($D_{p*}$), along with the geogrid, reduced the impact of nappe on the downstream bed, meaning that the scouring was significantly reduced, as compared to the NM cases during all $D_{c*}$ (0.1, 0.15 and 0.2).

The scouring phenomena observed during the G1M and G2M case were different compared to the NM cases because the combined effect of the water cushion and geogrid acts as a dual barrier, which reduces the impact of the approaching nappe flow. The maximum scour depth ($S_{d*}$) during G1M and G2M was mainly dependent upon the dimensionless aperture size ($d^*$) of the geogrid. The scour depth decreases with the increase in $D_{p*}$, as shown in Figures 6, 7 and 9a. Approximately 30% (during the NM case), 74% (during the G1M case) and 100% (during the G2M case) scour reduction occurred as the $D_{p*}$ increased from 0.30 to 0.45 at a constant $D_{c*}$ of 0.2 (Figure 9), while the scour depth increased with the increase in $D_{c*}$ during all G1M and G2M cases, as shown in Figures 6 and 7.

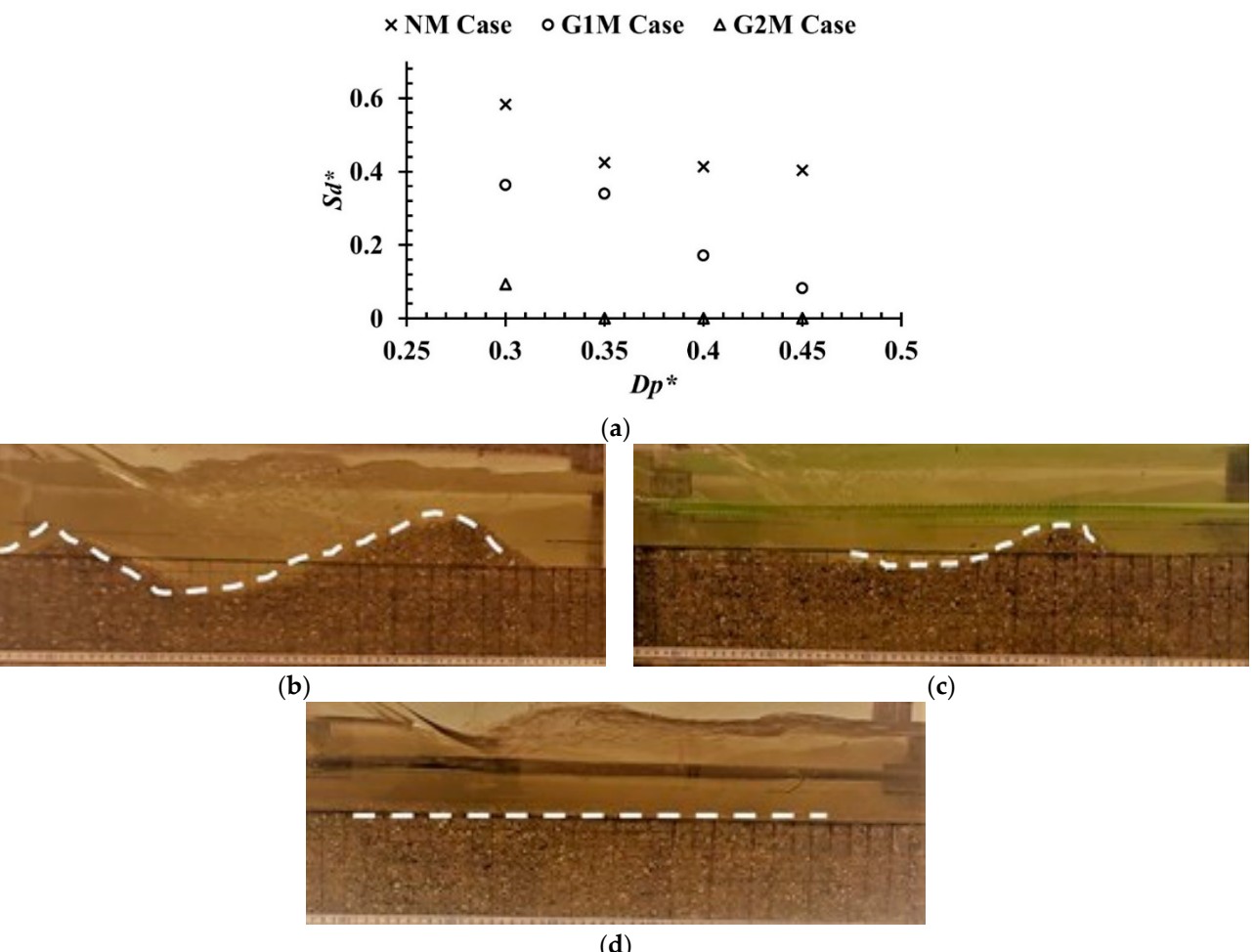

**Figure 9.** (**a**) Relation between maximum scour depth ($S_d$) and $D_{p*}$ at $D_{c*}$ = 0.2; pictorial view of scour profile (**b**) during NM Case ($D_{p*}$ = 0.45; $D_{c*}$ = 0.2); (**c**) G1M case ($D_{p*}$ = 0.45; $D_{c*}$ = 0.2); (**d**) G2M case ($D_{p*}$ = 0.45; $D_{c*}$ = 0.2).

The scour reduction during these cases (G1M and G2M) was mainly due to the presence of the geogrid, which suppresses the fluid force of the approaching nappe flow [35]. Although both geogrids performed very significant roles during the G1M and G2M cases as compared to the NM cases Figure 8, the performance of the G2M cases (G2 with aperture size $d$ = 2.5 mm) was more satisfactory than G1M (G1 with aperture size $d$ = 6.5 mm), as shown in Figure 8c,d. This was because 100% scour reduction was observed during the

G2M cases and was due to the aperture size of the geogrid. As the dimensionless aperture size ($d^*$) decreased ($d^* < 0.5$ during G2), the impact of nappe was significantly reduced by the finer openings of the geogrid, which significantly reduced the scouring. As the $d^*$ increased during G1 ($d^* > 0.5$), the impact of the approaching nappe was not significantly reduced due to the coarser aperture size, as compared to the G2. The difference between G1 and G2 can be observed in Figure 10.

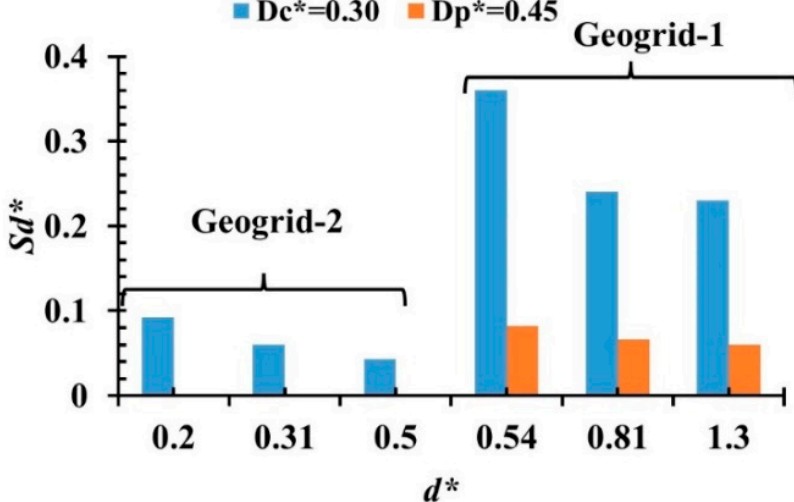

**Figure 10.** Relation between maximum scour depth ($S_d{}^*$) and dimensionless aperture size ($d^*$) at $D_{c^*} = 0.2$.

## 4. Conclusions

This study was conducted to introduce a novel countermeasure against scouring due to the direct collision of the overflowing nappe at the downstream side of the levee. Behind the levee, two countermeasure techniques were examined, i.e., by providing the water cushion/pooled water and the geogrids with two different aperture sizes, large (G1 = $d$ = 6.5 mm) and small (G2 = $d$ = 2.5). Flume experiments were conducted in a laboratory channel through two varying phases (rigid and moveable bed conditions). During the first phase (rigid bed cases), flow structure variations were observed, due to the formation of hydraulic jump by providing only dimensionless pooled water ranging from ($D_{p^*} = \textbf{0.3–0.45}$), named as (NR) no geogrid with rigid bed and then in combination with the selected two geogrids (G1R/G2R) against the three dimensionless overtopping depths ($D_{C^*} = \textbf{0.1, 0.15, 0.2}$). During the second phase (moveable bed material), the development of the scour process was observed by replacing the rigid bed with non-cohesive gravel bed material under the same conditions, i.e., varying the dimensionless pooled depths ($D_{P^*}$), geogrids with different aperture sizes (G1M/G2M) and the dimensionless overtopping depths ($D_{C^*}$). The following conclusions were derived from the present study.

When only pooled water ($D_{P^*}$) was applied behind the levee during both the first (rigid) and second (moveable) phase (i.e., NR and NM), the flow structures (hydraulic jump classification) and scour development significantly varied with the changing dimensionless pooled water depths ($D_{p^*}$) and overtopping depths ($D_{C^*}$). During the NM case, i.e., no geogrid with moveable bed, the scour depth increased by approximately 93% (when $D_{p^*} = 0.30$) and 62% (when $D_{p^*} = 0.45$) when the $D_{c^*}$ increased from 0.1 to 0.2. This means that the highest values of $D_{C^*}$ played a significant role in increasing the scouring at the downstream side of the levee, as compared to the lowest $D_{C^*}$ value. On the contrary, when the $D_{p^*}$ increased from 0.30–0.45, the scour depth decreased, i.e., around 17–31%.

The combination of the geogrid and pooled water (G1M and G2M cases) played a vital role in suppressing the scour depth. The performance of the G1M cases with aperture sizes $d = \textbf{6.5 mm}$ were more effective as compared to NM and approximately 57–78% scour reduction occurred during G1M after changing the $D_{p^*}$ from 0.30 to 0.45. On the contrary,

the G2M cases with aperture sizes $d = $ **2.5 mm** were more effective as compared to the G1M and NM cases because 100% scour reduction was reported during the G2M cases after the changing $D_{p^*}$ from 0.30 to 0.45. It was due to the dimensionless aperture size ($d^* < 0.5$ during geogrid 2) that the impact of nappe was significantly reduced by the finer openings of geogrid 2, which significantly reduced the scouring. Hence, the mesh size of the geogrid must be finer to suppress the scouring.

**Author Contributions:** Conceptualization, F.M.A. and N.T.; methodology, F.M.A. investigation, F.M.A.; writing—original draft preparation, F.M.A. writing—review and editing, F.M.A.; visualization, F.M.A. supervision, N.T. All authors have read and agreed to the published version of the manuscript.

**Funding:** This research received no external funding.

**Data Availability Statement:** Not applicable here.

**Acknowledgments:** The authors acknowledge the support of the MEXT scholarship (F.M.A.) from the Japanese Ministry of Education, Culture, Sports, Science, and Technology (Monbukagakusho). The authors also acknowledge the Amina for her assistance in conducting the experimental work.

**Conflicts of Interest:** The author declares no conflict of interest.

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
