# Peer review of "Utilization of Geogrid and Water Cushion to Reduce the Impact of Nappe Flow and Scouring on the Downstream Side of a Levee"

_fluids, doi:10.3390/fluids7090299_

Round 1

Reviewer 1 Report

The authors presented an experimental work with the aim to reduce the impact of overflowing nappe flow and associated scouring on downstream using various configurations.

The paper can be accepted after minor revision:

An actual photo of the experimental setup is to be added.

The title of Fig 3 is to be more specific (what is the configuration and what are the fixed parameters?)

It is better to use ‘’dimensionless’’ instead of ‘’ ‘’non-dimensional’’

An experimental uncertainty study is to be performed.

Information’s about the acquisition system are to be added.

Why haven’t you used fluorescence fluid for a better visualization of the flow?

The paper is to be checked against misprint and grammatical mistakes.

Author Response

Author’s Response to the Review Comments

Manuscript ID: fluids-1860884 R1

Title: Utilization of geogrid and water cushion to reduce the impact of nappe flow and scouring on the downstream side of a levee.

Corresponding Author: Prof. Dr. Norio Tanaka

All Authors: Fakhar Muhammad Abbas, Norio Tanaka*

We appreciate the opportunity to submit a revised version of our paper, so thank you very much. We are grateful to the editor and reviewers for their time and effort in evaluating this article. The comments of reviewer-1 have been addressed thoroughly in this file. For the sake of clarity, we've copied the reviewers' comments and responded to each one, along with the section and line number from the revised manuscript. It is our belief that we have addressed all the issues and that the updated version will meet the journal's requirements for publication.

Reviewer 1

The authors presented an experimental work with the aim to reduce the impact of overflowing nappe flow and associated scouring on downstream using various configurations.

The paper can be accepted after minor revision:

1-An actual photo of the experimental setup is to be added.

2-The title of Fig 3 is to be more specific (what is the configuration and what are the fixed parameters?)

3-It is better to use ‘’dimensionless’’ instead of ‘’ ‘’non-dimensional’’

4-An experimental uncertainty study is to be performed.

5-Information’s about the acquisition system are to be added.

6-Why haven’t you used fluorescence fluid for a better visualization of the flow?

7-The paper is to be checked against misprint and grammatical mistakes.

Response:

  • Thankyou for your kind comment. The picture of experimental setup has been added in the revised manuscript.
  • The correct information regarding figure has been revised. Line 220-221.
  • The word has been replaced with the mentioned word recommended by the kind reviewer.
  • The information regarding Uncertainty is now added in the revised manuscript.
  • Thankyou for your valuable comment. Now the information regarding the acquisition system has been added in the revised manuscript. Section 2.5.
  • Thankyou for your concern. We have used the aluminum powder (typically used to shine the water particles) in the water to visualize the flow. The information has now been added in the revised manuscript.
  • Thank you for your concern regarding this issue. Now the grammatical mistakes have been removed from the revised manuscript.

Reviewer 2 Report

The authors have to add a photo of the experimental setup.

Why haven’t you used a PIV system for the visualization?

An uncertainty study is mandatory

The acquisition system is to be presented with more details.

The conclusion is to be reduced.

Is the flow, laminar or turbulent?

The measurement techniques are not well described.

Where are placed the measurement sensors?

The resolutions of Figs 6 and 7 are to be improved.

More physical interpretations are to be added to the discussion

Have you checked the repetitiveness of the experimental results?

The authors have to explain how they got the analytical results presented in table 2.

Author Response

Author’s Response to the Review Comments

Manuscript ID: fluids-1860884 R1

Title: Utilization of geogrid and water cushion to reduce the impact of nappe flow and scouring on the downstream side of a levee.

Corresponding Author: Prof. Dr. Norio Tanaka

All Authors: Fakhar Muhammad Abbas, Norio Tanaka*

We appreciate the opportunity to submit a revised version of our paper, so thank you very much. We are grateful to the editor and reviewers for their time and effort in evaluating this article. The comments of reviewer-2 have been addressed thoroughly in this file. For the sake of clarity, we've copied the reviewers' comments and responded to each one, along with the section and line number from the revised manuscript. It is our belief that we have addressed all the issues and that the updated version will meet the journal's requirements for publication.

Reviewer 2

1-The authors have to add a photo of the experimental setup.

2-Why haven’t you used a PIV system for the visualization?

3-An uncertainty study is mandatory

4-The acquisition system is to be presented with more details.

5-The conclusion is to be reduced.

6-Is the flow, laminar or turbulent?

7-The measurement techniques are not well described.

8-Where are placed the measurement sensors?

9-The resolutions of Figs 6 and 7 are to be improved.

10-More physical interpretations are to be added to the discussion

11-Have you checked the repetitiveness of the experimental results?

12-The authors have to explain how they got the analytical results presented in table 2.

Response:

  • Thank you for your kind comment. The Figure has been added in the revised manuscript.
  • Thank you for your valuable comment. The flow was turbulent in the concerned area. Due to the limitations of the PIV system (Particularly regarding the turbulent flow), the flow cannot be visualized properly by the PIV system. Therefore, we used the high definition camera for flow visualization.
  • Thank you for your concern. Now the information regarding the uncertainty has been added in the revised manuscript. The measurements of the scour depths were obtained by positioning 3D laser displacement gauge just above the flume bed vertically at a small interval of 1-2 cm (for obtaining the accuracy) depending on the variation in the gravel bed both in the longitudinal and transverse directions. The 3D laser displacement gauge was connected with PC, and LJ navigator was used to get the data of scour profile which was further analyzed in Fortran Software to get the final values of scour depths.
  • Thank you for your kind comment. Now we briefly added the information regarding the acquisition system in the revised text. Line 234-237 & 247-249.
  • The conclusion has been reduced in the revised text. Line 483-517.
  • The flow was turbulent.
  • The section of measurement techniques has been improved in the revised text. Line 223-250.
  • Thank you for your concern. The details of the placement of measurement instruments have been added in the revised text. Line 234-237 & 247-249.
  • The resolution of Fig 6 and 7 has been improved in the revised text.
  • The discussion has been added in the revised text.
  • Yes, we have checked the repetitiveness of the results. There is no issue regarding this matter
  • We have mentioned in Table-2 from where we calculated the analytical values.

Reviewer 3 Report

The reviewed manuscript describes a method for studying the challenging problem of diminishing the impact of the overflowing nappe flow to which scouring at the downstream is associated. The problem if of a great importance, so the proposed research theme is relevant for many practical reasons. The proposed methodology is well conceived and the paper is clearly helping the reader to understand what is all about. Undoubtedly this is the merit of the authors. As far as I could see, the revised version of the manuscript makes it clearer than the initial form. The concluding remarks are supported by the reported results and they be accepted in the present form.

Overall, I consider the manuscript as publishable. However, there are still some drawbacks that need to be solved.  For instance, Figures 5-7 are difficult to read. I suggest the authors to change the colors of the graphs to avoid any confusion. Perhaps more contrasting colors will help the reader to distinguish easier the curves. Moreover, I recommend the authors to change the scale for the sour profile from [-10,7.5] as it is now in Figures 6 and 7 to [-6,6], thus making the plots readable. 

I wonder why all the variable appearing in the text are written in bold? Would the authors mean of changing them in regular type?

Last, but not the list, I suggest one more careful read since some minor linguistic problems are still persistent.

Author Response

Author’s Response to the Review Comments

Manuscript ID: fluids-1860884 R1

Title: Utilization of geogrid and water cushion to reduce the impact of nappe flow and scouring on the downstream side of a levee.

Corresponding Author: Prof. Dr. Norio Tanaka

All Authors: Fakhar Muhammad Abbas, Norio Tanaka*

We appreciate the opportunity to submit a revised version of our paper, so thank you very much. We are grateful to the editor and reviewers for their time and effort in evaluating this article. The comments of reviewer-3 have been addressed thoroughly in this file. For the sake of clarity, we've copied the reviewers' comments and responded to each one, along with the section and line number from the revised manuscript. It is our belief that we have addressed all the issues and that the updated version will meet the journal's requirements for publication.

Reviewer 3

The reviewed manuscript describes a method for studying the challenging problem of diminishing the impact of the overflowing nappe flow to which scouring at the downstream is associated. The problem if of a great importance, so the proposed research theme is relevant for many practical reasons. The proposed methodology is well conceived, and the paper is clearly helping the reader to understand what is all about. Undoubtedly this is the merit of the authors. As far as I could see, the revised version of the manuscript makes it clearer than the initial form. The concluding remarks are supported by the reported results, and they be accepted in the present form.

Overall, I consider the manuscript as publishable. However, there are still some drawbacks that need to be solved.  For instance, Figures 5-7 are difficult to read. I suggest the authors to change the colors of the graphs to avoid any confusion. Perhaps more contrasting colors will help the reader to distinguish easier the curves. Moreover, I recommend the authors to change the scale for the sour profile from [-10,7.5] as it is now in Figures 6 and 7 to [-6,6], thus making the plots readable. 

I wonder why all the variable appearing in the text are written in bold? Would the authors mean of changing them in regular type?

Last, but not the list, I suggest one more careful read since some minor linguistic problems are still persistent.

Response:

Thank you for your valuable comments on our revised manuscript. According to suggestion of reviewer Figure 5-7 have been revised with more clear and thick color curves. Now the curves are very clear and distinguishable.

The scale of Figure 5-7 has also been revised.

The variable used in the word format was not bold, but when we paste in Fluid Journal template it automatically generated with bold format.

However, regarding the minor revisions in this round, we now have resolved all the minor issues regarding the Figure’s clarity and linguistic problems in the latest revised version.

Reviewer 4 Report

The reviewed article concerns the utilization of geogrid and water cushion to reduce the impact of nappe flow and scouring on the downstream side of a levee. In this study, laboratory experiments were conducted with the three cases for rigid bed (R) named as NR, G1R, G2R (N, G1 and G2 represent no geogrid, geogrid-1 and 2, respectively), and moveable bed (M) named as NM, G1M, 17 G2M, to elucidate the effect of dimensionless pooled water depths (DP*), overtopping depth (DC*) and aperture size of geogrid (d*) on flow structure and scouring. The subject matter seems interesting. The assumed research goal has been achieved. The subject of the article is in line with the profile of the journal.

Author Response

Author’s Response to the Review Comments

Manuscript ID: fluids-1860884 R1

Title: Utilization of geogrid and water cushion to reduce the impact of nappe flow and scouring on the downstream side of a levee.

Corresponding Author: Prof. Dr. Norio Tanaka

All Authors: Fakhar Muhammad Abbas, Norio Tanaka*

We appreciate the opportunity to submit a revised version of our paper, so thank you very much. We are grateful to the editor and reviewers for their time and effort in evaluating this article. The comment of reviewer-4 has been addressed in this file. It is our belief that we have addressed all the issues and that the updated version will meet the journal's requirements for publication.

Reviewer 4

The reviewed article concerns the utilization of geogrid and water cushion to reduce the impact of nappe flow and scouring on the downstream side of a levee. In this study, laboratory experiments were conducted with the three cases for rigid bed (R) named as NR, G1R, G2R (N, G1 and G2 represent no geogrid, geogrid-1 and 2, respectively), and moveable bed (M) named as NM, G1M, 17 G2M, to elucidate the effect of dimensionless pooled water depths (DP*), overtopping depth (DC*) and aperture size of geogrid (d*) on flow structure and scouring. The subject matter seems interesting. The assumed research goal has been achieved. The subject of the article is in line with the profile of the journal.

Response:

Thank you for your positive and valuable appreciation.

Reviewer 5 Report

This study focuses on characterizing the scouring depth and length due to nappe flow, and how this scouring varies with overflow depth and geogrid from experimental and analytical point of view. The introduction and analysis is interesting. I think this manuscript is acceptable after minor revision.

Specific points:

1.      It seems the manuscript is still under editing, for example, several colors of the texts could be found, pictures in figure 2 starts from b

2.      It’s difficult to identify the four lines in Figure 7.

3.      The difference between G1M and G2M could be added in the abstract.

4.      Table 4 could be reproduced with less lines and clear ways. This is because for different Dp*, flow structure classification is the same.

Author Response

Author’s Response to the Review Comments

Manuscript ID: fluids-1860884 R1

Title: Utilization of geogrid and water cushion to reduce the impact of nappe flow and scouring on the downstream side of a levee.

Corresponding Author: Prof. Dr. Norio Tanaka

All Authors: Fakhar Muhammad Abbas, Norio Tanaka*

We appreciate the opportunity to submit a revised version of our paper, so thank you very much. We are grateful to the editor and reviewers for their time and effort in evaluating this article. The comments of reviewer-5 have been addressed thoroughly in this file. For the sake of clarity, we've copied the reviewers' comments and responded to each one, along with the section and line number from the revised manuscript. It is our belief that we have addressed all the issues and that the updated version will meet the journal's requirements for publication.

Reviewer 5

This study focuses on characterizing the scouring depth and length due to nappe flow, and how this scouring varies with overflow depth and geogrid from experimental and analytical point of view. The introduction and analysis is interesting. I think this manuscript is acceptable after minor revision.

Specific points:

  1. It seems the manuscript is still under editing, for example, several colors of the texts could be found, pictures in figure 2 starts from b
  2. It’s difficult to identify the four lines in Figure 7.
  3. The difference between G1M and G2M could be added in the abstract.
  4. Table 4 could be reproduced with less lines and clear ways. This is because for different Dp*, flow structure classification is the same.

Response:

Thankyou so much to the kind reviewer for careful review of our manuscript.

  1. Thank you for indicating this mistake. The several colors in the manuscript text were due to track changes because the file submitted was a revised version of the original manuscript. We have also revised the caption of Figure 2 in the revised file.
  2. Thank you for your valuable comment. Actually, the data in the cases show similar values/trends by changing the Dp* therefore lines were overlapping each other.
  3. The description of G1M and G2M is now added in the abstract.
  4. Thank you for your kind comment. Now Table 3 has again been revised.
